# Spent Brewer’s Yeast Lysis Enables a *Best Out of Waste* Approach in the Beer Industry

**DOI:** 10.3390/ijms252312655

**Published:** 2024-11-25

**Authors:** Livia Teodora Ciobanu, Diana Constantinescu-Aruxandei, Ileana Cornelia Farcasanu, Florin Oancea

**Affiliations:** 1Bioproducts Group, Bioresources Department, National Institute for Research & Development in Chemistry and Petrochemistry–ICECHIM, Spl. Independentei No. 202, Sector 6, 060021 Bucharest, Romania; livia.ciobanu@icechim.ro; 2Interdisciplinary School of Doctoral Studies ISDS–UB, University of Bucharest, Bd. Mihail Kogalniceanu No. 36-46, 050107 Bucharest, Romania; ileana.farcasanu@chimie.unibuc.ro; 3Faculty of Biotechnologies, University of Agronomic Sciences and Veterinary Medicine of Bucharest, Bd. Mărăști No. 59, Sector 1, 011464 Bucharest, Romania

**Keywords:** yeast cell walls, glucan, chitin, elicitor, mannoprotein, emulsifier, yeast extract, antioxidant activity, taste enhancers

## Abstract

Yeasts have emerged as an important resource of bioactive compounds, proteins and peptides, polysaccharides and oligosaccharides, vitamin B, and polyphenols. Hundreds of thousands of tons of spent brewer’s yeast with great biological value are produced globally by breweries every year. Hence, streamlining the practical application processes of the bioactive compounds recovered could close a loop in an important bioeconomy value-chain. Cell lysis is a crucial step in the recovery of bioactive compounds such as (glyco)proteins, vitamins, and polysaccharides from yeasts. Besides the soluble intracellular content rich in bioactive molecules, which is released by cell lysis, the yeast cell walls β-glucan, chitin, and mannoproteins present properties that make them good candidates for various applications such as functional food ingredients, dietary supplements, or plant biostimulants. This literature study provides an overview of the lysis methods used to valorize spent brewer’s yeast. The content of yeast extracts and yeast cell walls resulting from cellular disruption of spent brewer’s yeast are discussed in correlation with the biological activities of these fractions and resulting applications. This review highlights the need for a deeper investigation of molecular mechanisms to unleash the potential of spent brewer’s yeast extracts and cell walls to become an important source for a variety of bioactive compounds.

## 1. Introduction

Microorganism diversity offers a spectrum of highly significant biomolecules. The bioactive potential of these compounds has been revolutionizing the biotechnological fields of pharmaceutical, food, and agricultural industries. The way in which these bioactive molecule factories are manipulated in favor of life-quality progress leads the way in sustainably exploiting the macro-benefits of this micro-world [1,2].

Yeast has been used in fermenting foods and beverages for a long time. Nevertheless, the biological constitution of yeast cells revealed that there is more to yeast than just a fermenting tool. Yeast cells are plentiful in valuable polysaccharides (β-glucans, mannans, chitin), proteins, peptides, and amino acids. Spent brewer’s yeast stands out also through its elevated content of micronutrients (group B vitamins, minerals) and hop-derived polyphenols [3,4].

The key step in valorizing yeast bioactive compounds is cellular disruption (lysis), which broadly refers to the breaking of the cell wall. This process allows the recovery and extraction of biomolecules from the cells. This review will extensively detail the main cell lysis methods used for spent brewer’s yeast (SBY), which is the main by-product of the beer industry. Considering an average of 1.9 billion hectoliters of beer produced worldwide annually and a range of 1.7–2.3 g of residual SBY resulting from each liter of beer, estimations for SBY global production per year vary between approximatively 323,000 to 437,000 tons [5,6,7,8]. As waste management is becoming an increasingly important issue nowadays, making use of by-products that are produced in high amounts and have such great nutraceutical value is imperative [9,10]. SBY is not only a source of nutrients but also a source of functional ingredients, including bioactive peptides with antihypertensive, antioxidant, and antimicrobial activities [11] with applications in the food, biotechnological, and pharmaceutical industries [7]; antioxidants with applications in cosmetics [12]; emulsifiers for edible oil microencapsulation [13]; β-glucan used as a thickener, and prebiotic in the food industry [14] or for skincare applications [15].

To the best of our knowledge, this is the first survey that reviews the lysis methods of SBY cells and the relation of these methods to the biological activities of the resulting products. The advantages and disadvantages of each lysis technique and also the most relevant experimental parameters which might accommodate further reproduction of these processes at a laboratory or industrial scale are discussed. Characterization of the resulting yeast-based products–yeast extracts (YEs) and yeast cell walls (YCWs) in terms of bioactive compounds is highlighted for each technique. This study also provides an overview of the properties and potential applications of YCWs, which is still an underutilized yeast derivative, despite the fact that is just as easily obtainable as YEs. Different applications of YE are connected to the biological activities of the main components. The need for the development of a complex quality control system to compensate for the high SBY variability and to deliver reproductible products is highlighted. In the end, a network bibliometric analysis of the main keywords is done to depict the main trends in the scientific literature regarding spent brewer’s yeast cell lysis.

## 2. Spent Brewer’s Yeast Composition

There are two main types of brewer’s yeast, with many varieties. Most brewer’s yeasts are included in the genus *Saccharomyces.* These yeasts are traditionally divided into two groups: *ale* yeasts (those specific to British beers) and *lager* yeasts (those specific to German beers). These groups are also known as top-fermented and bottom-fermented yeasts because they were initially classified based on their flocculating properties. At the end of fermentation, the upper yeast tends to rise to the surface of the fermented wort/malt extract, while the lower yeast settles to the bottom of the fermentation vessel [16]. Brewer’s yeasts of *ale* type are *S. cerevisiae* [17]. *Lager* yeasts are *S. pastorianus* (synonymous name *S. carlbengensis*), a hybrid of *S. cerevisiae* with *S. eubayanus*, with two types, type I and type II [18]. Yeasts from other genera than *Saccharomyces* have a significant innovation potential in the beer industry [19]. However, the utilization of non-*Saccharomyces* yeast is still limited, despite their advantages in terms of beer quality [20,21]. Therefore, in this review, we will focus mainly on spent brewer’s yeast from the genus *Saccharomyces*.

SBY was exhaustively characterized predominantly with regard to its nutritive content. The elevated protein content of this biomass has been ubiquitously recognized, accounting for approximatively 40–65% of the dry matter. The second most abundant compounds in SBY are carbohydrates. The data on this matter seem to vary depending on the type of biomass studied and its provenience. Values as low as 12.9% were reported, while other assays mentioned a range of 22 to 54% carbohydrates in SBY. Yeast cell walls are a consistent source of polysaccharides such as glucans and chitin. Their structure is detailed in Section 4 of this review [7,22,23,24].

The remarkable nutritive profile of SBY also includes micronutrients, mainly B-group vitamins and minerals. Among the B-group vitamins, nicotinic acid (B3) and pyridoxine (B6) are the most prevalent with content as high as 104 and 55.1 mg/100 g dry weight, respectively. While SBY also contains a wide variety of minerals, sodium, magnesium, and potassium seem to be the most abundant in this category [7,24].

A particular trait of this microorganism, which has been only recently discussed, is the capacity of yeasts to adsorb the polyphenols which are released by hops used in the beer production and fermentation processes. This leads to an increase in the antioxidant activity of SBY [22,25].

## 3. Spent Brewer’s Yeast Cell Lysis Methods

Cell lysis, also known as cellular disruption, is the process of breaking the cell wall and plasma membrane with the aim of releasing the intracellular content of the cell (e.g., organelles, biomolecules such as proteins, DNA, RNA, polysaccharides, etc.). The cellular disruption methods can be broadly classified as non-mechanical and mechanical.

Non-mechanical methods (autolysis, enzymatic hydrolysis, thermolysis, solvent treatments, etc.) can be subdivided into three main categories—physical, chemical, and biological—and are based on diverse principles. These methods are milder and more specific than mechanical methods because they are designed to perforate the cells rather than completely disintegrate them. Certain interactions between substances and cell wall components can permeabilize the cell membranes, leading to intracellular compound release. Non-mechanical methods are typically low cost and readily reproductible. Chemical reaction wastes and the low potential of industrial scaling are some of the main cumbersome issues of non-mechanical methods [26,27,28].

Mechanical cell lysis methods (ultrasonication, bead milling, high-pressure homogenization, etc.) involve the use of external forces, commonly shear forces, to physically break the cell membrane. These methods are recognized especially for their high effectiveness, suitability for various cell types, and also for their high throughput [26,28,29]. Another important advantage is that they do not require further disposal of any hazardous chemical residues. Moreover, the processes are usually automatized, and the experiments are usually performed using specialized equipment. However, the mechanical disruption methods require high energy input and might not be fit for extraction of certain molecules (e.g., shear-sensitive biomolecules such as DNA) [27,30].

Choosing the suitable disruption method depends mainly on the type of cell, as well as on the aim of the experiment (e.g., extraction of a particular intracellular biomolecule/compound). This part of the review focuses on the methods that are intensively used for the lysis of yeast cells from spent brewer’s yeast biomass. Figure 1 displays a schematization of these methods.

### 3.1. Non-Mechanical Lysis Methods

#### 3.1.1. Autolysis

Autolysis is the irreversible self-degradation of the cell by endogenous enzymes (glucanases, mannases, and chitinases). The consequences of this event are the loss of permeability of the plasmatic membrane and altered porosity of the cell wall, which eventually leads to the release of intracellular compounds. In yeast cells, the dissolution of the organelles might be causing the release of enzymes such as proteases and carbohydrases from vacuoles into the cytoplasm. Autolysis can be a natural phenomenon in yeast, taking place at the end of the stationary cell growth phase due to the natural aging of cells, or it can be induced by physiological stress of a chemical or physical nature (variations of temperature–thermolysis, pH imbalance, osmotic pressure caused by plasmolyzing agents–NaCl, acids, etc.) [31,32,33]. Induced autolysis of yeast cells has become a valuable biotechnological process due to its wide applications in industries that use yeast-based products. Autolysis of yeast cells is considered a mild disruption method and, therefore, allows the extraction and recovery of various intracellular/cell wall yeast compounds with bioactive or nutritional potential, such as proteins, amino acids, polyphenols, and polysaccharides, with minimal or no risks of biocompound alteration [34,35]. However, the prolonged time of reaction and the addition of potential chemicals to fix this issue could represent disadvantages when applying this lysis method [32].

Autolysis is among the most utilized methods for spent brewer’s yeast cell lysis. Most of the processes described in the literature mention 15–18% yeast suspensions autolyzed at temperatures between 45–60 °C within a time range of 16–30 h. Quantifying the compounds that are released in the YE is a useful method of evaluating the lysis efficiency, as the quantity of the released compounds correlates with the lysis efficiency. The result overview confirmed the peak of the intracellular component release at an optimum 45–50 °C temperature [36,37]. Generally, the main components of the YE are proteins and carbohydrates, with overall concentration ranges between 45–55% and 13–30%, respectively [36,37,38,39,40,41,42,43].

Tanguler et al. showed that temperatures above 55 °C do not improve autolysis [15]. The solid, protein, and α-amino nitrogen contents of the autolysate obtained at 60 °C were lower than those obtained through autolysis at lower temperatures of 45°, 50°, or 55°. In their experiments, the yields of intracellular compound release reached a plateau after 24 h of autolysis. Given its nutritional value, integrating YE in food recipes came as a natural attempt. Hence, their study also carried out the sensory analysis of the autolysate, which was added in different concentrations (0.5–2.0% *w*/*v*) to vegetable soup. However, the panelists preferred the soups with lower YE concentrations [36].

There are several studies that address the characterization of SBY autolysates from a nutritional perspective, highlighting the potential of yeast components in functional foods and dietary supplements. For example, Tessaro et al. used dried and concentrated SBY autolysate as an ingredient in cookies [38]. Świderski et al. focused on the analysis of SBY autolysates with high amino-acid content [39]. They obtained YE with 77.5% amino acid content. This was a particular case, as the raw material used was spent brewer’s yeast from wort suspension (which is rich in proteins, peptides, and free amino acids). The amino acid profile showed that the most prevalent amino acids of the autolysate were phenylalanine, tyrosine, leucine, valine, and lysine, which are known to fulfill important biological functions. Moreover, the polyphenol content of the extract was relatively high (228.3–336.1 mg GAE/100 mL), thus accounting for the high antioxidant activity of the SBY autolysates [39]. Other studies also confirmed that leucine, valine, and lysine are the main amino acids found in SBY extracts [40,44]. The studies of Oliveira et al. measured also the mineral content of the autolysate. The results showed a total content of 36.9 ng/g minerals, among which phosphorus and potassium were the most abundant [40].

Saksinchai et al. successfully integrated concentrated autolysate in growth media for *Bacillus thuringiensis*. The slurry suspension of 15% (*w*/*v*) solid content was autolyzed for 20 h at 50 °C. The YE had a relatively high α-amino nitrogen content compared to other studies, 4.5% (*w*/*w*), which made it fit as an ingredient for the growth media [41]. The study also suggests that YE could promote the growth of certain bacteria and that certain yeast-based compounds could exhibit prebiotic activities.

Autolysis can also be used in the extraction of β-glucan or mannoprotein from YCWs. Thanardkit et al. obtained yeast cell walls with 57% (*w*/*w*) carbohydrate content and 29% (*w*/*w*) crude protein content after 20 h autolysis at an optimum temperature of 50 °C. β-glucan was further isolated from other carbohydrates by alkaline treatment with 1 M NaOH [42]. For mannoprotein recovery, an optimum 8 h autolysis at 37 °C, using a phosphate buffer, was used to obtain yeast cell walls from brewery waste biomass. The cell walls were treated with NaOH and sedimented with ethanol for mannoprotein precipitation. The final content of mannoprotein was 44.85% [45]. Still, it is essential to point out that the lysis method is just a preliminary step in these kinds of experiments and that the subsequent extraction methods substantially influence the extraction rates. In some cases, chemical methods are applied in addition to autolysis to enhance cell disruption. Similarly to alkali treatments, acids can be used to weaken and permeabilize the cell wall and plasma membrane. Although these methods stand out through their effectiveness, toxic compounds, which might form during this process, greatly restrict further utilization of the treated biomass. Moreover, these harsh treatments might disintegrate certain nutritive compounds like amino acids [46,47]. In the case of SBY, articles describe the use of sulfuric or clorhidric acid to disrupt the yeast cells. While Heringer et al. reported no significant differences, in terms of protein release, between acid hydrolysis (with HCl) and enzymatic hydrolysis or plasmolysis [48], Alves et al. reported increases of 38.63% and 55.83% in total solid and protein extraction, respectively, when sulfuric acid was added at the beginning of the autolysis reaction [32].

A summing-up of the main autolysis parameters for SBY and experimental results regarding the content of the resulting YEs (autolysates) and YCWs is provided in Table 1. Although each study reported the content of different compounds, depending on their focus, the data still provides a useful overview of the highly nutritional value of YEs and YCWs which are rich in proteins, carbohydrates, and essential amino acids (AA).

#### 3.1.2. High-Temperature Lysis

While mild thermal treatment is used to induce autolysis through endogenous enzyme activation, high temperatures can exhibit disruptive effects on yeast cells as well. Cells exposed to high temperatures or heat shocks usually suffer an increase in intracellular osmolarity, which induces osmotic pressure that contributes to the cell wall weakening and consequent outburst [49,50]. Moreover, thermolysis is known to cause molecular rearrangements in the plasma membrane, increasing its fluidity, while concomitantly degrading the bonds between the cell wall components, which also results in cell rupture [51,52,53].

However, this technique should be adapted to the experimental purpose, i.e., the targeted biomolecules for recovery, since high temperatures can alter thermolabile compounds with valuable bioactive potential [51,52]. Although this method finds useful applications in the extraction of thermostable enzymes from bacterial cells, its applications in SBY cell lysis seem to be limited, considering the lack of data on this topic [30,54]. This is probably due to the fact that high temperatures treatments might diminish the nutraceutical value of the resulting SBY derivatives, as a result of the thermic degradation of some compounds with biological activity.

An innovative approach to SBY valorization using high-temperature treatment came from Thakkar et al. by the implementation of a hydrothermal treatment method for yeast cell lysis, known as *flash hydrolysis*. The biomass was exposed to temperatures ranging from 160 °C to 280 °C, for only 10–12 s, the core objective being nutrient recovery for subsequent use for bacterial cell growth. Yeast and the remaining nitrogen and carbon residues resulting after SBY processing at an optimum of 240 °C proved to be an ecological alternative to the reagent-consuming microbiological media for the efficient growth of *Escherichia coli* [55]. Nevertheless, in addition to the elevated processing temperatures, the pressure applied through the equipment system also represents a contributing factor to cell disruption of yeast cells in this case [56].

Heringer et al. proposed a heat treatment method for SBY cell lysis as a means of autoclavation. A 0.25% SBY solution was autoclaved at 115 °C for 10 min, obtaining 60.2 ± 0.12 g/100 g^−1^ soluble protein, 10.38 ± 0.02 g/100 g^−1^ total nitrogen, and 33.48 ± 0.08 g/100 g^−1^ total organic carbon in the SBY lysate. Even if this heat treatment was the least efficient method in comparison to the other lysis methods described (acid hydrolysis, plasmolysis, autolysis, and enzymatic hydrolysis), the statistical assay showed only marginal differences between results [48].

#### 3.1.3. Plasmolysis

Cells placed in hypertonic solutions, where the extracellular environment has a higher concentration of solute in comparison to the intracellular space, react by the excessive release of water in an attempt to preserve the osmotic balance. As the cytosolic volume decreases, the cell will shrivel, causing ruptures in the plasma membrane which leads to the release of the soluble intracellular content [57,58].

The cellular mechanisms involved in maintaining the osmotic gradient in hypertonic conditions rely on the intracellular accumulation of some low-molecular compounds (solutes), namely osmolytes. This response prevents water efflux from the cell and thus, maintains the homeostatic cell volume. In plasmolysis, these compounds are manipulated in favor of cell disruption to create an opposite effect of osmotic equilibrium disturbance [59,60]. Specifically, resuspending cells in osmolyte solutions provide the necessary hypertonic environment which impels plasmolysis. Studies report the use of both organic (sucrose, trehalose, or organic solvents such as ethyl acetate, toluene, acetone, or ethanol) and inorganic osmolytes (salts such as NaCl, KCl as sources of sodium, potassium, and chlorine ions) in plasmolysis [60,61,62].

Plasmolysis was reported to be just as efficient as autolysis and enzymatic lysis in terms of the soluble protein content of the lysates. Heringer et al. report 63.08 ± 0.15 g soluble proteins/100 g^−1^ in the lysate resulting after SBY cell plasmolysis with ethyl acetate. The total nitrogen content (10.90 ± 0.3 g/100 g^−1^) was higher in the samples processed in such a manner in comparison to acid lysis, autolysis, thermal treatment, or enzymatic lysis, while the total organic carbon content (34.75 ± 0.75 g/100 g^−1^) was only marginally different in comparison to the aforementioned lysis methods described in this study. This data confirms that plasmolysates are a good source of nutrients for microbial growth media, as the authors have reported that their study focused on using SBY processed through different lysis methods for *Lasiodiplodia theobromae* MMPI fungus cultivation and production of β-D-glucan exopolysaccharide (lasiodiplodan) [48].

Disruption of SBY cells through plasmolysis is also emerging as an efficient micronutrient biosorption method. Costa et al. have managed to enrich SBY biomass with vitamin D3 (cholecalciferol) after plasmolysis with 10% NaCl at 55 °C for 48 h. The resulting biomass was mixed with a hydroalcoholic solution of cholecalciferol and stirred for various amounts of time, obtaining an optimum of 12 h and 15 h processing time. The impregnation and retention efficiency were 35.94% and 12.3% higher in the plasmolyzed biomass in comparison to the intact yeast cells [63].

In spite of its effectiveness, plasmolysis is limited by the high content of the resulting processed biomass or the need for additional purifying/washing steps for chemical removal [60].

#### 3.1.4. Pulsed Electric Field Lysis

Electric pulses of short duration (10^−4^–10^−8^ s) and high voltage (5–50 kV/cm) can be used to disrupt cells [64,65]. The most probable explanation involves the capacity of membrane phospholipids to act as electric dipoles while encountering an external electric field, that imposes a transmembrane potential greater than the rupture potential. As the transmembrane potential is formed, phospholipidic rearrangements occur leading to pore formation, which is known as electroporation or electropermeabilization [66,67]. A pulsed electrical field (PEF) is highly suitable for the extraction of both thermostable and thermolabile or sensitive compounds. Therefore, it can be used for the isolation of proteins, saccharides, and even antioxidants or nucleic acids [68].

This technique was only relatively recently used to disrupt yeast cells from brewery wastes. Liu et al. present an optimization of pulsed electric field SBY cells lysis, highlighting the important influence of experimental parameters such as electric field strength, pulse duration, and liquid-solid ratio. At a 50:1 liquid-to-biomass ratio and 8 µs pulses of 10 kV/cm voltage, an optimum of 2.788 ± 0.014% of protein extraction yield was achieved [69].

PEF lysis on SBY can also be performed as a pre-treatment for extraction of different bioactive molecules such as amino acids, glutathione, and proteins, as Berzosa et al. have described. Their study confirms that an electric field of at least 10–12 kV/cm has to be applied to induce electroporation in SBY suspensions. Even more, an acidic pH of 3.5–4 facilitates PEF lysis. The maximum content of amino acids, glutathione, and proteins were 162.52 mg/g dw, 2.94 mg/g dw, and 382.27 mg/g dw, respectively, but these results were dependent mostly on the extraction methods which were used after PEF cell disruption [70].

#### 3.1.5. Enzymatic Hydrolysis

Enzymatic hydrolysis (EH) is the lytic decomposition of the cell wall using exogenous enzymes. Such lysing agents can be enzymes of microbial origin (e.g., bacteriolytic enzymes such as glycosidases, acetylmuramoyl-L-alanine amidases, and endopeptidases). Several microorganisms, such as *Cytophaga* spp., *Rhizoctonia* spp., and *Cellulosimicrobium cellulans* (former names are *Oerskovia xanthineolytica* and *Arthrobacter luteus*) were reported to produce enzymes with yeast-lysing potential [71]. Compared to other cell lysis methods, enzymatic hydrolysis is considered mild in terms of preserving the intracellular released compounds and it is suitable especially for obtaining high-purity proteins [30,72,73]. Nevertheless, the main disadvantage of this method refers to the high cost of enzymatic reagents involved in the process [28].

Considering the rigid structure of the YCW, an efficient hydrolysis relies on the capacity of the enzymes to exhibit sequential and synergistic lytic effects. Specifically, the sequential effects refer to the ability of the enzymes to successively attack and disrupt the cell wall layers, i.e., the inner layer of β-1,3 and β-1,6 glucan branches and the outer mannan-protein matrix. The combined use of enzymes with specific activity (e.g., glucanases, proteases, etc.) can constitute a synergistic effect, due to their ability to disintegrate the different layers of the yeast cell wall [74]. Therefore, the enzymatic hydrolysis of the YCW requires a mix of different enzymes containing β-1-3-glucanase, protease, β-1-6-glucanase, mannanase, and chitinase, to exert the aforementioned synergic effect. These mixtures are available on the market either for laboratory or industrial uses. The effect of the blend of enzymes which is required for the hydrolysis of the YCW involves mainly three steps: the decomposition of the mannan-protein outer layer under protease action, followed by the glucanase attack of the glucan inner layer and, consequently, the cell burst (releasing intracellular components) due to the osmotic pressure induced by the weakening of the cell wall structures [30,74,75].

Most of the studies of EH using SBY were performed using yeast suspensions with approximately 5–12% biomass concentrations. The process is simple, requiring only the addition of enzyme, usually in relatively low concentrations, to the yeast sample and continuous stirring at a proper temperature. The optimum temperature that was mentioned in almost all of the studies was 50 °C, regardless of the enzyme type. At this temperature, the action of the lytic enzyme reaches its maximum efficiency. The pH is another crucial parameter for EH, as the optimum pH range depends on the type of enzyme that is being used in the process. For the most part, a commercial enzymatic mixture was used to perform EH of SBY. The most cited enzymatic mixtures were commercially available endo-protease mixes—Protamex^®^, Alcalase^®^, Neutrase^®^–and endo-peptidase mixes–Flavourzyme^®^.

Bayarjargal et al. used Flavourzyme^®^ (pH = 8.0) and pancreatin (pH = 6.0) to obtain SBY hydrolysates. Comparative experiments demonstrated that the lysis with pancreatin was the most efficient in terms of the amount of released intracellular compounds. After 5 h of hydrolysis of a 10% (*w*/*v*) yeast suspension with 2.5% (*w*/*v*) concentration of pancreatin, the resulted yeast hydrolysate had 55.9% protein content and 4.8% α-amino nitrogen [37].

A study that focused on valorizing SBY by creating yeast hydrolysates with high CHP (Cyclo-His-Pro) content—a bioactive molecule that has been associated with glycemic control in diabetes—compared the action of five enzymes: Neutrase^®^, Alcalase^®^, Protamex^®^, Flavourzyme^®^ and fincin (1:100 ratio enzyme:yeast substrate). The results of this study showed that the Flavourzyme^®^ treated hydrolysate had the highest CHP content, so further analysis was conducted on this extract of interest. After 48 h of hydrolysis, the crude protein content was 64.9% and the carbohydrate content was 26.9%. The amino acid profile was also examined: the most abundant amino acids were glutamic acid, leucine, and lysine. The hydrolysate also exhibited high antioxidant activity [76].

Increasing the EH time leads to elevated protein yield. The protein content was measured after 6 h, 12 h, and 24 h of EH with Alcalase^®^. After 24 h of hydrolysis, the protein content doubled. Yet, when the quantity of enzyme used in the reaction was doubled, the amount of protein in the extract was not significantly higher [77].

Sequential EH using different enzymatic mixtures is an approach that could maximize both the recovery of the released products and the antioxidant properties of the YE produced from spent brewery yeast. This was confirmed by the assay of a hydrolysate obtained through a Brauzyn^®^ treatment (a vegetable protease, active on yeast cell walls) that was furtherly subjected to an Alcalase^®^ treatment [78].

Most of the previously described surveys assessed mainly the YE composition. Tam et. al. focused on the YCW composition, as the goal was β-glucan extraction. Alcalase ^®^ 2.4 LFG was used for the EH of SBY, and a 15% (*w*/*w*) slurry suspension in phosphate-citrate buffer (pH = 7.0) was hydrolyzed for β-glucan extraction. The optimum enzyme concentration was 0.86% (*w*/*w*) E:S. After 5 h of hydrolysis, autoclavation, and organic solvent treatment (for β-glucan extraction), the YCWs had a 23.33% protein content and a relatively high content of β-glucan [79].

The content of the previously described yeast enzymatic hydrolysates, with a focus on their nutritional valuable compounds (when reported), are included in Table 2, along with the main parameters and enzymes/enzyme blends used in the EH process.

### 3.2. Mechanical Lysis Methods

Considering the number of publications on the topic, there seems to be a preference for non-mechanical methods against mechanical methods, when considering the single-technique lysis of SBY. Numerous studies chose to combine mechanical methods with other non-mechanical/mechanical techniques to enhance the effectiveness of the process. Such approaches will be detailed in Section 3.3 of this study.

#### 3.2.1. Ultrasonication

Exposing cells to ultrasounds (≥20 kHz frequency) is often used as a yeast cell lysis method. The mechanism of cell disruption through ultrasonication is based on the cavitation phenomenon. Cavitation refers to the formation, growth, and collapse of vapor/gas bubbles (cavities) within a liquid medium. In ultrasonication processing, high acoustic power leads to microbubble formation in liquid samples. For a better understanding of the ultrasonication mechanism, it is important to state that sounds create longitudinal wave oscillations. When sound propagates through liquids, it generates high-pressured, compressed areas and also low-pressured, stretched areas of the molecular spacing. These two phases are called compressions and rarefactions, respectively [80,81,82]. The rarefaction phase allows the growth of the microbubbles, while the compression phase presses the bubbles, leading to their disintegration. This breakage induces shock waves that are transmitted through the liquid sample. During the ultrasonication process, these waves are responsible for continuously weakening the cell walls until their disruption takes place [83].

It is difficult to provide a very precise overview of the general optimum parameters of ultrasonication in SBY, as the reported experimental parameters and the analysis methods used for evaluating cell disruption efficiency vary greatly. However, most of the studies report that ultrasonication for 20–30 min, at 25–30 °C, using highly diluted SBY suspensions (<10%) and sample volumes of 200–250 mL leads to an effective batch cell lysis [40,43,44,79].

One of the surveys regarding the ultrasonication of SBY took aim at determining the amino acid profiles of YE obtained through ultrasonication. After 30 min. of sonotrode ultrasonication (20 kHz) at 7 °C, with 400 W power input, the lysis effectiveness was reported to be 80%. With regard to the amino acid content, the predominate amino acids were glutamate, aspartate, alanine, and arginine. Taking into account the nutritional value, the highest amount of essential amino acids was reported for leucine, lysine, and valine [44]. Oliveira et al. processed a 1% yeast suspension via ultrasonic probe, for 30 min. using 60 s pulses and 15 s off period, obtaining 50.2% maximum protein content of the YE [40]. Other surveys attempted β-glucan extraction and determined the YCW protein and β-glucan content which was reported 26.3% and 56.5%, respectively. These results were obtained after ultrasonication of 15% (*w*/*w*) yeast suspension, at a power input of 8.29 W/g for 11.6 min. The extraction yield was 42.55% [79].

A rather different approach was designed by Bertolo et al.: an ultrasonic bath with 40 kHz input was used for ultrasonication, for 1 h at 30 °C. The lysed yeast was directly dried after the process and the protein and carbohydrate content was 42.83 g/100 g of dry basis and 53.91 g/100 g of dry basis, respectively [43].

From our personal experience in performing ultrasonication using spent brewer’s yeast, the yield of the process is inversely related to the yeast sample concentration and to temperature [84]. It is also essential to adapt the sample volume to the capacity of the probe, which is usually specified in the device manual. The time of ultrasonication, as well as power input, might be important factors, but considering the energy that is transmitted into the system is more relevant. Moreover, the SBY biodiversity highly influences the biological activity of the yeast extract resulting from ultrasonication [85]. Overall, as long as over-heating is avoided, targeted compounds for extraction through ultrasonication remain intact. Therefore, protein purification, nucleic acid, or lipid extractions are easily achievable using this method [86].

#### 3.2.2. Bead Milling (Beating)

Disruption of cell walls through bead milling is based on the shear forces generated by the rotary movement of beads and cells in the grinding chamber of bead mill machines and also by the forced bead-cell collisions promoted by the same rotary movement [87,88]. Bead beating, which refers to rapidly stirring the sample with grinding media (beads) using certain devices that are not necessarily bead mills, follows the same principle, disrupting the cells through agitation, friction, and collision. Depending on the biomass type, glass, zirconium, or steel beads can be used. An efficient disruption depends on choosing the right type of beads (density, diameter) in correlation with the biomass type and concentration, the optimum bead-cell ratio, flow rate, agitation speed, etc. [89,90,91]. It is important to state that, just like other mechanical methods (e.g., high-pressure homogenization), bead milling could imply the use of expensive tools and equipment and could also generate excessive heat which might alter thermolabile compounds [28].

Caballero-Córdoba et al. provided a nutritional evaluation of a YE obtained through bead milling disruption of spent brewer’s yeast cells. The 40% (*w*/*v*) yeast cell suspension was subjected to bead mill lysis at a 2400 rev min^−1^ rotation rate and 4.8 l h^−1^ flow duty, and 70% of the mill grinding chamber was filled with 0.6 mm glass beads [92]. The cell lysis was efficient, occurring in over 95% of cells. This was confirmed also by the high release rates of intracellular compounds. The protein content of the YE reached 78%, and the carbohydrate and lipid content were 21.52% and 3.53%, respectively. Concerning the amino acids, phenylalanine, tyrosine, lysine, leucine, threonine, and valine prevailed. These results were in accordance with most of the previously discussed studies which examined the same compounds in the SBY extracts [92].

An even higher disruption efficiency was obtained by Avramia et al. [93]. Their β-glucan extraction from SBY cell walls implied bead cell lysis as a primary step. Different concentrations of yeast cell suspensions were added in tubes that were 85% filled with glass beads. The yeast suspension to glass beads ratio and the number of 10 min. vortexing cycles were also varied to optimize the process. The highest disruption efficiency was 99.8% and was achieved using a 5% yeast suspension, at a 1:2 yeast/glass bead ratio and 3 vortexing cycles [93].

A potential application in the food industry of a YE that was harvested after bead-beating disruption was proposed by Pancrazio et al. [94]. Their survey assessed the effect of incorporating 1% of SBY extract into cooked hams. To disrupt the yeast cells, samples of 1:2:1 biomass:acetate buffer:glass beads were vortexed 10 times for 1 min. each. The compounds with nutritional value within the YE were quantified, specifically proteins (1.1 g/100 mL), lipids (<0.02 g/100 mL), and amino acids (glutamine, lysine, serine, and valine were higher than 100 µg/mL). There were no major changes regarding the sensory analysis of the ham, or its physical properties. Increased hardness and chewiness were reported, due to the stronger gel formed during cooking, which was maintained during the storage of the ham. Thus, the YE could be used as a gel stabilizer for cooked ham, consequently enhancing its processing conditions [94].

#### 3.2.3. High-Pressure Homogenization

High-pressure homogenization (HPH) emerged as a technology primarily used for emulsion preparations, through nano/micro-fragmentation of samples. Nonetheless, this system proved just as efficient for cell disruption and has been used for many years now. The exact mechanism of cell lysis through high-pressure homogenization still needs further clarification, but it is thought to be contingent on a combination of physical procedures such as cavitation, turbulence, impingement, pressure oscillations, and shear forces induced during these processes [95,96]. Broadly, the basis of the cell disruption mechanism by way of HPH refers to a pump that is forcing the sample through an extremely narrow orifice of a valve system, where pressure is rapidly delivered. In addition to the resulting alternation of cell compression (while entering the valve orifice)–cell expansion (while leaving the valve orifice), shear forces are also produced through the impact of the cells with the solid surfaces of the valve system [28,30,97,98,99]. Regardless of HPH efficiency and scalability, equipment costs and potential heat-induced damage of extracted compounds should be taken into consideration [28].

To the best of our knowledge, studies that used HPH as a single-method for disruption of spent brewer’s yeast cells are extremely limited. One of the most relevant ones was the experiment described by Oliveira et al., which compared the efficiency of several lysis methods using SBY, including HPH. The processing of a 39% (*w*/*v*) yeast suspension required 5 passes through the machinery system at pressures between 600–1000 Ba. The resulting YE had a protein content between 44.5 and 51.2%, depending on the yeast strain that was used, which suggested that, at least in regards to the protein release, the HPH lysis was almost as efficient as ultrasonication, more efficient than enzymatic hydrolysis, but less efficient than autolysis. It is important to emphasize that HPH is a highly scalable process [40]. HPH seems to enhance the lysis of SBY when used as part of a multiple-step process. This subject is described in the next subsection of this review.

### 3.3. Cascade and Combined Lysis Experiments for Disruption of Spent Brewer’s Yeast Cells

One of the strategies that can be employed for streamlining the cell lysis process is designing cascade experiments that combine two or more disruption methods. By all means, the experimental variables depend on the type of sample and on the desired result. This subsection offers some insight into this matter by providing some examples where combined lysis methods were applied to disrupt the spent brewer’s yeast cells.

Liu et al. presented a multiple-step cell lysis experiment, with the final aim of extracting β-glucan from spent brewer’s yeast cell walls [100]. The main steps were autolysis, hot water treatment, high-pressure homogenization, organic solvent treatment, and enzymatic hydrolysis through protease treatment. The first results showed that autolysis of a 15% (*w*/*w*) solid content slurry, at 55 °C, for 24 h, caused the release of both proteins and carbohydrates, but at different rates. Even if the protein release was much slower in comparison with carbohydrate release, the final protein concentration was higher than the carbohydrate concentration in the YE. This outcome is also confirmed by the results discussed in the 2.2.1. section of this review. Autolysis was also promoted by the use of NaCl 3% as a plasmolyzing agent. The hot water treatment, optimized at 4 h of autoclavation at 121 °C, was efficient for removing most of the mannoproteins, which was advantageous for the further extraction of β-glucan. Heating of the sample also seemed to have facilitated the cell disruption. After 3 passes through a high-pressure homogenizer at 700 Ba, the rate of disrupted cells was 95.5%. Suspending the obtained cell walls in organic solvents was necessary for removing most of the lipids. Protamex was ultimately used for removing any mannoprotein residues and its action increased the β-glucan content of the processed biomass to 93.12% [100].

Combining HPH with other disruption techniques can increase the β-glucan content of YCWs. This was shown by Thammakiti et al., who performed autolysis of 15% (*w*/*w*) SBY cells under standard conditions, at 50 °C, for 24 h. The autolyzed yeast cells were passed 6 times through a high-pressure homogenizer at 600 Ba. This process increased the polysaccharide content (from 65% to 72%) and the β-glucan content (from 55% to 59%). Furthermore, the authors suggested that using autolysis along with HPH might enhance the water-holding and stabilizing capacities of the obtained β-glucan biomass [101]. Other studies indicate that using HPH might increase both the quantity and the purity of the compounds extracted from spent brewer’s yeast. For example, the study from Tian et al. group proved that the NaOH alkaline treatment of yeast cell walls obtained after the centrifugation of previously SBY suspension subjected to HPH (1700 Ba, 2 passes) yields high purity β-glucan. The system could be also used on an industrial scale [102].

Barbieru et al. developed a method for SBY cell lysis based on HPH and enzymatic hydrolysis, taking aim at maximizing protein extraction. Using specialized software, an experimental design was developed to analyze and optimize yeast concentration, homogenizer pressure, number of passes, and enzyme concentration. The most efficient lysis was obtained using 10% yeast concentration, 1000 Ba pressure, 3 passes, and 0.1 g/L VinoTastePro enzyme. The protein content of the YE was 11.22 mg/mL. Moreover, their experiments proved that intensifying the process (e.g., increasing the sample concentration or the number of passes) does not boost the lysis process [103]. Their results are in agreement with Liu et al. [100].

An interesting approach to valorizing SBY was put forward by Sombutyanuchit et al. [104]. Preparation of 5′-GMP-rich YE was achieved through autolysis followed by enzymatic hydrolysis with 5′-phosphodiesterase extracted from malt rootlet. The target molecule of the experiment–5′-GMP–can be used to enhance the yeast extract flavor, which is one of the reasons that an increase in its concentration is desired. The YE resulting after 8–20 h of autolysis with 5′-phosphodiesterase indicates that extended autolysis is associated with a decline in the 5′-GMP content of the YE. The 5′-GMP levels increase proportionally with the phosphodiesterase level but are inversely related to the enzymatic hydrolysis time [104].

Some studies proved that exploiting the nutritional values of SBY hydrolysates can be improved through ultrafiltration and nanofiltration. SBY was autolyzed (4 h, at 70 °C) and then ultrafiltered with a cut-off of 10 kDa [105]. Both of the resulting fractions (retentate and permeate) were subjected to hydrolysis using 4% plant extract from *Cyanara cardunculus*. The hydrolyzed fractions were nanofiltered with a 3 kDa cut-off. The high molecular weight fractions (>3 kDa) were rich in proteins, and the low molecular fractions (<3 kDa) were rich in carbohydrates, small peptides, amino acids, and minerals (especially sodium and potassium). The separation and concentration of the SBY lysis product through filtration lead to obtaining fractions with dietary and bioactive potential [105].

In another study, the same authors showed that fractions < 3 kDa, obtained through similar steps, contain high amounts of angiotensin-converting enzyme (ACE)-inhibitory peptides, that are stable in the in vitro simulated gastrointestinal system, highlighting their potential applications in hypertension therapeutics [106].

As the lysis strategies are highly dependent on the type of compounds that are targeted for extractions, the correlation between the aim of the experiment and the multi-step lysis is presented in Table 3.

## 4. Bioactive Compounds from Yeast Cell Walls

Surveys that portray cell lysis as a method of SBY valorization are mostly focused on yeast-soluble extracts, their characteristics, and their uses. Still, much remains to be investigated about the yeast cell wall applications, as their great importance represents a hot field of scientific research.

Yeast cell walls are known for their strength and their high content of bioactive compounds. According to most studies, the cell wall of yeasts has 60% β-glucan, 16–40% mannan (of which 80–90% mannose and 10–20% mannoproteins), and 1–2% chitin. These percents can vary greatly, depending on the yeast strain, growth stage, and conditions [107,108,109,110,111]. Polysaccharides and proteins of the YCW matrix are tangled, and yet harmoniously organized. The YCW architecture can be widely described as a fibrillar network of β-1,3-glucan linked to a β-1,6-glucan branched structure through trans-glycosidic bonds, which are successively trans-glycosylated with chitin and GPI (glycosylphosphatidylinositol)–anchored mannoproteins. The proteins are O-linked and N-linked to mannose moieties, creating a mannan polymer [112,113]. The whole structure is kept together by covalent interactions. Mannoproteins are covalently crossed-linked to β-1,3-glucan either indirectly through β-1,6-glucan structures, or directly through disulfide, glyco esther bonds, or hydroxyl groups of β-1,3-glucan, etc. [112,114].

The yeast cell wall composition offers a wide spectrum of applications in food, agricultural, biomedical, and pharmaceutical industries, e.g., thickener for dairy products [14], elicitor of plant innate immunity [115] adjuvant of immune response antitumoral agents or promotor of osteogenesis [116], a component of skincare formulations [15], coatings for pharmaceutical products [117], etc. The number of publications concerning this topic has increased over the last years, as the need for sustainable resources and the development of novel therapeutics are becoming an urgent topic.

The majority of the available publications regarding the applications of YCW describe the uses of YCW components (β-glucan, mannoproteins, and chitin). The value of the YCW components is indisputable. However, the use of pure β-glucan, mannoproteins, or chitin from the YCW structure demands a series of extraction steps in addition to cell lysis. These isolation processes involve the use of chemicals and are generally resource-consuming. Using pure whole YCW directly from yeast biomass could be tremendously advantageous, as it would not require extraction procedures. Hence, this section addresses the valuable potential of yeast cell walls, YCW components, and their possible applications, with a focus on valorizing pure YCWs.

### 4.1. Applications and Properties of YCW Components

#### 4.1.1. Yeast β-Glucan Properties and Applications

An impressive range of studies present the bioactive properties of β-glucan and specifically of yeast β-glucan, which is considered to have the highest bioactivity compared to other sources (mushrooms, fungi, etc.). Petravić-Tominac et al. provided a data overview on this topic [118]. Their survey detailed the most prevalent biological effects of yeast β-glucan. The medical uses of this compound as an immunomodulator, adjuvant in cancer therapy, infection prophylactic, and tissue regeneration enhancer were discussed, with exemplification of relevant results. Topics regarding the use of yeast β-glucan in skin care products were tackled. Applications that are less often mentioned in other publications were introduced (e.g., β-glucan from yeast used in the preparation of construction products like concrete/mortar, in laboratory processes as a support for enzyme immobilization and solid support in chromatographic separation methods, etc.) [118]. Immunomodulation, which is the main characteristic of β-glucan has been already comprehensively reviewed [116,119,120].

Applications of yeast β-glucan as a food ingredient are also a topic that has been already studied and reviewed. Zechner-Krpan et al. highlight the properties that make this compound a good food additive in addition to its reputable health benefits: β-glucan from SBY can be used as a thickener, water holding, and oil binding agent and also as a gelifier [121]. More recent reviews which covered this subject emphasized the use of β-glucan as a fat replacer ant stabilizer for mayonnaise, prebiotic and health benefit enhancer in yogurt, and as a source of fiber and glycemic control in bread [122]. A review that was published recently described the applications of β-glucan in the food industry in association with its biological effects. Among many other studies on β-glucan, Caruso et al. provided an overview of a large number of studies which show the prebiotic, immunomodulating, antitumoral, and hypocholesterolemic properties of this compound when administered or fed to fish, shrimps, piglets, broilers, or mice [123].

#### 4.1.2. Yeast Mannoprotein-Properties and Applications

Few studies described the applications of yeast mannoproteins (MPs), but the existing publications point out mainly the potential of these compounds in the food industry. Mannoproteins have stabilizing and emulsifying properties. Whether they were integrated into mayonnaise or salad dressings, MPs demonstrated the prior properties without altering the sensorial characteristics of the food. It was observed that the emulsifying effects were comparable to those of gum arabic or xanthan gum and lecithin, which are very efficient commercial emulsifiers [124,125,126,127]. MPs can be also used as a stabilizer in wines, while also enhancing the wine quality and taste [128,129].

The stabilizing features could extend to more various domains of applications, for example stabilizing biologically active compounds (e.g., anthocyanidins, nanoparticles). Jine Wu et al. outlined the ability of mannoprotein to bind to anthocyanins through hydrophobic interactions, forming stabilizing complexes. This formulation can preserve the anthocyanin state when exposed to thermal stress, reducing the degradation rate, while the antioxidant activity and color are conserved [130]. Maghemite (γ-Fe_2_O_3_) nanoparticles were incorporated within mannoproteins, which evidenced the stabilizing properties of mannoproteins [131].

Antifungal activities against *Aspergillus flavus* in pistachio were described by Abdolshahi et. al. [132]. The pathogen growth was greatly inhibited in pistachios that were coated with MPs from *S. cerevisiae* yeast cell walls [133]. MP antibacterial activity against enteropathogens such as *Campylobacter jejune*, *Escherichia coli*, and *Salmonella* sp. were mentioned. Moreover, the probiotic capacities of MPs were noticed, as they contributed to the survival of beneficial lactic acid bacteria in (simulated) gastrointestinal juice, improving their growth rate and their viability in the gastrointestinal tract [132].

#### 4.1.3. Yeast Chitin–Properties and Applications

Research on chitin biopolymers of natural origin has exponentially expanded over the last few years. Chitin is biocompatible, biodegradable, non-toxic, and sustainable. On top of that, chitin has a high biological activity. Just like β-glucan, chitin has antimicrobial, antioxidant, and wound-healing properties. Low molecular chitin is thought to be the most biologically active form of chitin. Some studies point out its capacity to lower blood sugars, triglycerides, and cholesterol, which suggests that it could be used in therapeutic strategies for metabolic diseases. Chitin could be a valuable resource for many industries, such as food, biotechnology, pharmaceutical, biomedical, materials industries, etc. Not to mention that recent studies on this compound described it as a potential tool in cancer diagnostic or gene therapy. All these properties and applications have already been thoroughly reviewed [134,135,136].

However, to the best of our knowledge, a specific overview of yeast chitin applications has not been provided yet. This might be also due to the lack of studies in this particular area. The available data on these subjects reveal the beneficial effect of chitin on plant health, as chitin can be an efficient antifungal agent [137].

There seems to be a particular interest in chitin–glucan complexes from yeast. Some studies propose strategies for the synthesis of these complexes, usually using *Pichia pastoris* yeast and crude glycerol as its main carbon source [138,139]. This interest might emerge also from the possibility of integrating chitin-glucan complexes into hydrogels, which are highly compatible formulas with promising potential in biomedical, biomaterial, and crop performance fields [140,141].

#### 4.1.4. Potential Applications of Yeast Cell Walls

A compelling application of YCW from SBY as a pharmaceutical additive was brought forth by a series of connected articles by Kasai et al. and Yuasa et al. [117,142,143]. The main idea of the study was to use acid-treated YCW formulas as coating agents for certain drugs. The intracellular components of the yeast cells were solubilized and released through intracellular/extracellular enzymatic hydrolysis, and the supernatant of the sample which contained these components was removed. The pellet which contained the YCWs was furtherly treated with HCl at high temperature (80 °C), resulting in acid-treated yeast cell walls. Coating acetaminophen tablets with this product was performed, revealing some interesting kinetic behavior regarding the release of the active ingredient under these conditions. The graphic representation of acetaminophen release was sigmoidal, which suggested that there was an incipient lag in the release rate of acetaminophen [117]. The studies also showed that higher curing time and temperatures can increase the lag time in the release of acetaminophen [142]. When acid-treated YCWs were sprayed on acetaminophen provided in the form of granules, a rapid release of the active ingredient was observed. Thus, YCWs could improve drug delivery systems by allowing control over the release time of drugs [143].

The use of plastic, especially in food packaging, is a universally recognized issue, as its disposal can raise serious environmental concerns. Additionally, plastics contain BPA (bisphenol-A), which is thought to migrate into food, especially at high temperatures. Consequently, finding alternatives to plastics constitutes a great need [144,145]. A study suggested that YCW waste from the brewery processes can be of great use in this matter as YCWs and glycerol formulas could be used for biodegradable food-contact materials fabrication [146].

YCWs have beneficial effects on plant health as well. YCW-based products were commercialized mainly as plant biostimulants. Plant biostimulants are a class of products applied for the treatment of cultivated plants that are defined by their agronomical functions, i.e., increased nutrient uptake and use efficiency, enhanced plant tolerance to stress, and improved yield quality [147]. One of the categories of plant biostimulants are elicitors [148], compounds that activate the plant defense mechanism following their perception by pattern-receptor [149]

YCWs are a source of oligosaccharins, respectively, compounds produced by the hydrolysis of polysaccharides that act as a signal for activating a plant’s defense response [150] due to their pattern being similar to the microbe-associated molecular pattern, MAMP. One of these active ingredients produced from YCWs, called cerevisane, modifies the expression of genes after application to a grapevine [151]. Overexpressed genes are those coding for the plant’s response to stress: (i) proteins involved in a plant’s defense processes (i.e., pathogenesis-associated proteins, phenylalanine-ammonia-lyase, stilben synthase, lipoxygenase, protein kinase; protein receptors with a high leucine content, non-specific plant lipid transfer proteins, serine-threon proteins involved in signal transduction, superoxide dismutase and glutathione S-transferase involved in response to oxidative stress); (ii) enzymes involved in the metabolism of hormones associated with the defense system (salicylic acid, jasmonic acid, ethylene); (iii) secondary metabolites (in particular polyphenols) and (iv) photosynthetic processes (chlorophyll A/B and related proteins, and components of light harvesting photosystems). For this type of biostimulant, commercialized under the name Romeo^®^, the EP1965649 (B1) patent (Lesaffre–Agrauxine, Beaucouze, France) was granted [152].

Another commercial product based on YCWs is called Hosaku Monogatari (HM) and is marketed by Asahi (Sumida, Tokyo, Japan), as a plant activator, i.e., as a compound that triggers and enhances the defensive response of plants against pathogens (an elicitor of the plant defense mechanism, similar to plant biostimulants). HM was produced from SBY, *S*. *pastorianus*, by enzymatic treatment (with protease YL-15, Amano Enzyme, Nagoya, Japan) followed by centrifugation. Narusaka et al. described the antibacterial and antifungal action of HM when applied to different plant species (*Arabidopsis thaliana*, *Brassica rapa*) against pathogenic *Pseudomonas syringae* and *Colletotrichum higginsianum* [115]. Antibacterial action of YCWs against *Penicillium expansum* was noticed in post-harvested pear fruit. It is considered that YCWs might modulate the expression of defense-related genes [153].

YCWs might also act as a mycotoxin adsorbent. Integrating them into the diet of certain animals, such as aflatoxin-affected broilers, proved to be a relatively efficient therapeutic and prophylactic strategy [154,155]. The potential applications and bioactive effects of YCWs are detailed in Table 4.

## 5. Yeast Extract

Yeast extract, i.e., the content of yeast cells, has several applications that are based on various activities of the components, nucleotides, proteins, amino acids, sugars, and different oligo- and micro-elements [156]. One of the applications is as a taste enhancer and an alternative to monosodium glutamate [157]. Yeast extract (hydrolysate) is an alternative to monosodium glutamate (MSG) as it contains not only *umami* peptides but also compounds that act synergistically with these *umami* peptides and (ribo)nucleotides, respectively, 5′-inosinic acid (5′-IMP) and 5′-guanylic acid (5′-GMP) [158]. Beyond these *umami* peptides, yeast extract also contains other types of peptides, responsible for the *kokumi* taste, which are also considered (a high-level) alternative to MSG. *Kokumi*, respectively, is “the richness of taste” and implies a taste sensation of a powerful intensity that is maintained for a long time. The scientific terms used are “prolonged continuity”, “density”, and “consistency” of taste [159]. The taste sensation of *kokumi* was first highlighted in an aqueous garlic extract in a *umami* solution [160]. The most active compounds of *kokumi* are γ-glutamyl peptides, including reduced glutathione and γ-Glu-Cys-Gly–GSH [161,162]. Peptides undergoing a Maillard reaction (MRP) also taste *kokumi* [163,164]. The Maillard reaction is a reaction between the carbonyl groups of different sugars and the amino groups of amino acids/peptides [165], which lead to the formation of compounds involved in the taste and aroma of heat-treated products [164,166] and those that have gone through slow fermentation processes [167,168]. It should be noted that, although the media periodically present *kokumi* as a new taste, *kokumi* peptides do not have specific taste receptors (like the other five tastes, sweet, sour, bitter, salty, and umami), but enhance the perception of these five other tastes. *Kokumi* peptides interact with the calcium-sensing receptor (CaSR) from the taste cells [169]. These kokumi peptides, due to their function as enhancers of salty and umami tastes, have significant health-promoting potential. Both tastes are essential for tasty food. Salt reduction is essential for hypertension control [170]. However, the saltiness taste is essential to enjoy food and poor patient adherence to salt reduction compromises blood pressure control [171]. *Kokumi* γ-Glutamyl peptides could support “salt reduction without reducing saltiness” and could improve the hypertension control [172]. Reduction of the MSG consumption due to its enhancement of umami taste has also benefic health effects due to the dual face of glutamate [173]. Yeast extract is an important source of *kokumi* peptides. Recently two peptides have been reported, IQGFK and EDFFVR, with *kokumi* activity [174].

For the production of yeast extract used as a taste enhancer, the yeast produced by aerobic cultivation (called in everyday language baker’s yeast) is preferred, not SBY resulting from an anaerobic/fermentation process. However, SBY contains more biologically active compounds, such as glutathione [10]. Glutathione is a tripeptide that generates the *kokumi* taste [161]. This high glutathione content allows the development of unique tastes and aromas—e.g., due to its high glutathione content, brewer’s yeast extract is a product from which the “beef” flavor can be more easily obtained, further enhancing the umami taste [31]. Several reasons hamper SBY’s use in the production of taste enhancers. One is related to the already mentioned high variability of SBY. Another reason relates to the bitter compounds like hops used to flavor beer adsorbed on SBY. A combined process of cell lysis by high-pressure homogenization followed by separation by microfiltration of the cell walls fails to produce yeast extract without bitter compounds. It was found that, although the bitter taste was partially removed by separating the cell walls from the extract, this separation was not total since the cell lysis at high pressure also caused partial desorption from the yeast cell wall and solubilization in the yeast extract of the compounds responsible for the bitter taste [175].

Developing a quality assurance control system is essential for managing reproducibility/non-compliance risks associated with scaling up SBY lysis processes for the production of taste enhancers based on yeast extract. Such a system involves both feed-forward control (to identify the particular characteristics of the raw material, resulting from its high variability) and a feed-back control (based on the characteristics of the finished product). An interphase control system is also required. Developing these types of controls involves correlation with specific methods to identify the factors essential for the quality of the product—Figure 2.

The yeast extract treatment with glutaminase could increase the content of *γ*-glutamyl peptide [176]. Drum drying of yeast extract promotes the formation of Maillard peptide and kokumi enhancers [162].

## 6. Network Bibliometric Analysis of the Main Keywords

To depict the main trends in the scientific literature regarding spent brewer’s yeast cell lysis, we have used the VosViewer bibliometric analysis tool. The sources cited in the current paper were fed into the VosViewer software, version 1.6.20 to generate a keyword co-occurrence network map based on text data. The terms were extracted from title and abstract fields, using the default binary counting option. The minimum number of connections between terms was set by 2 terms. The Network Visualization assay (Figure 3) provided a network map divided into 7 clusters, illustrated as differently colored groups of circles.

The clusters represent groups of related keywords. Moreover, the size of each circle is positively correlated with the occurrence of its stated keyword in the analyzed text corpus, i.e., title and abstract. Also, the distance between the circles is negatively correlated with the co-occurrence of the terms in the provided text data. The yellow cluster, containing 5 items—bioactive molecule, microorganism, fermentation, review, and application–points out that a range of studies described the products derived from microorganisms or fermentation processes with regard to their bioactive molecule content. The term “application” stands out, which suggests that the studies focused on exploiting these compounds with bioactive value. The cluster depicted in blue shows that the terms extraction, cell disruption, optimization, and comparison are often associated. This illustrates that “cell disruption” is often associated with “extraction”, i.e., of certain compounds. The elevated size of the “extraction” circle in comparison to the other circles from this cluster confirms that cell lysis is a commonly used method for the valorization of yeast-derived compounds. Therefore, it is only natural that some researchers provided data on optimization and comparison of cell disruption methods. The green-colored cluster clearly shows that there is a great interest in mannoproteins present in the cell walls of yeasts. This term seems to be frequently associated with the word “production”, which might suggest a tendency toward describing mannoprotein isolation. The red cluster encompasses some terms directly related to yeast cell walls (e.g., cell walls, chitin, glucan). The association of these keywords with “enzyme” most likely refers to the enzymatic lysis processes. Spent brewer’s yeast is also included in this cluster. The size of this circle is, however, considerably smaller, and connected mainly to the glucan content. This information calls attention to the reduced number of studies of this underestimated by-product and, implicitly, the novelty of the topic. Interpretation of these maps in association with the overlay visualization (described below), estimates that this research endeavor has only emerged over the last few years.

The overlay visualization (provides the visualization of the main keywords extracted from the title and abstract fields of the sources cited in the current paper in relation to the publication year—Figure 4.

The scale states the publication time period and its corresponding color. The overlay network map demonstrates that yeasts have been intensely studied in the 2010s with regards to their cell wall content (mannoproteins, chitin, glucan), uses in the food industry, or even cell disruption methods and their optimization. However, the tendencies slowly transitioned towards the bioactive molecule content of spent yeast derivatives, their extraction, and valorization, which seem to have gradually transpired as hot topics over the last 4 years.

## 7. Conclusions

Spent brewer’s yeast is a brewery by-product produced in high quantities., which remains underutilized and is disposed of despite its valuable bioactivity and its potential applications in a wide range of fields such as food and agricultural sectors, the cosmetic industry, and biomedical sciences. This review described the main methods used for SBY lysis, which could provide an accessible alternative for yeast biomass valorization and sustainable brewery waste management. This work provided an overview of the mechanism and main experimental parameters of SBY lysis methods, as well as of the bioactive compounds content of the resulted yeast derivatives–YEs and YCWs–in correlation with their applications with great innovative potential. Owing to high prevalence, non-toxicity, and ease of manipulation, it is expected that components recovered from the lysis of spent brewer’s yeast will have significant contributions as a part of the solution for several of today’s challenges, such as safer taste enhancers, more efficient formulation of pharmaceuticals, nutraceuticals, fertilizers, and plant biostimulants as well as safer and more sustainable biodegradable plastics materials.

## Figures and Tables

**Figure 1 ijms-25-12655-f001:**
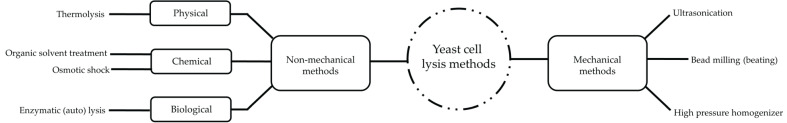
Yeast cell lysis methods. Classification of cell lysis methods that are most commonly used for spent brewer’s yeast processing.

**Figure 2 ijms-25-12655-f002:**
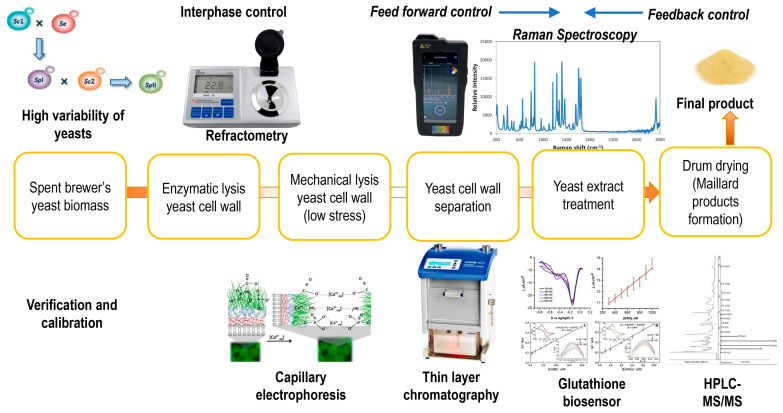
Illustration of the quality insurance system intended to compensate for the high variability of the raw material, the spent brewer’s yeast. This system also involves using a feed-forward system designed to adapt the parameters of the enzymatic-mechanical lysis technology to reduce bitter compound desorption from the yeast cell walls. The high-pressure liquid chromatography with mass spectrometry detector (HPLC-MS/MS) certifies the presence of kokumi peptides and supports a robust feedback (control) system.

**Figure 3 ijms-25-12655-f003:**
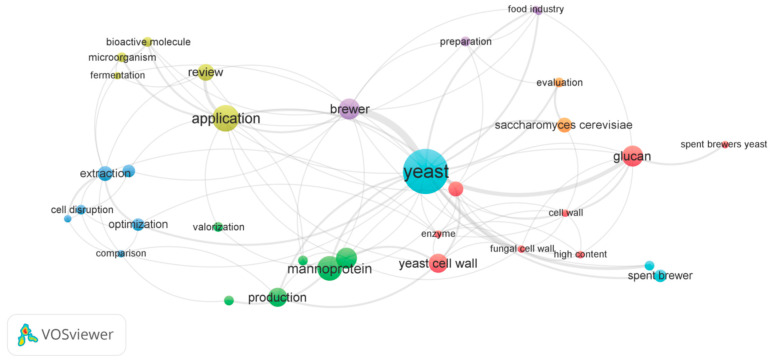
Network Visualization of the keywords co-occurrence.

**Figure 4 ijms-25-12655-f004:**
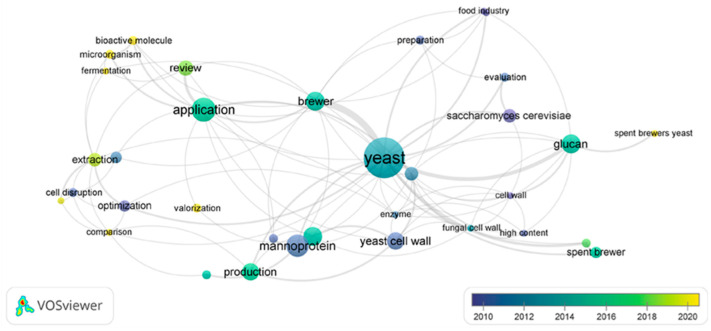
Overlay visualization of the keywords co-occurrence.

**Table 1 ijms-25-12655-t001:** Main autolysis parameters and nutritional content of different yeast fractions obtained through autolysis of SBY.

Yeast Concentration (% *w*/*v*)	Temperature (°C)	Time (h)	Analyzed Fraction	Content (% *w*/*w*)	Reference
15	50	24	YE	Proteins: 48.70% Carbohydrates: 13.9% α-amino nitrogen: 3.9%	Tanguler et al., 2008 [36]
50	55	24	YE	Proteins: 45.30% Carbohydrates: 29.36% Lipids: 0.97%	Tessaro et al., 2020 [38]
15.5–16.5 *	47	48	YE	Proteins: 7.53% Amino acids: 77.5% Main AA: Phe + Tyr, Val, Leu + Nle Polyphenols: 228.3–336.1 mg GAE/100 mL	Swiderski et al., 2015 [39]
7 *	50	24	YE	Main AA: Leu, Lys, Val, Thr	Jacob et al., 2019 [44]
50	50	24	YE	Proteins: 54.50% Carbohydrates: 14.8% Main AA: Leu, Val, Lys, Thr Main minerals: P, K	Oliveira et al., 2022 [40]
15	50	20	YE	Amino nitrogen: 4.5% Carbohydrate (as glucose): 26.8%	Saksinchai et al., 2001 [41]
18 *	50	20	YCW	Proteins: 29% Carbohydrates: 57%	Thanardkit et al., 2022 [42]
50	55	24	whole autolyzed yeast	Proteins: 39.32% Carbohydrates: 46.48% Lipids: 1.25%	Bertolo et al., 2019 [43]

* Concentrations marked with ‘*’ are expressed ‘on a dry basis’. The other concentrations express the amount of yeast pellet obtained by centrifugation of pre-treated (washed) yeast biomass. The yeast species was *Saccharomyces cerevisiae* in all cases.

**Table 2 ijms-25-12655-t002:** Main parameters of enzymatic hydrolysis (EH) and nutritional content of different yeast fractions obtained through EH of SBY.

Yeast Sample Concentration (% *w*/*v*)	Temperature (°C)	Time (h)	Enzyme/ Enzyme Mix	Enzyme Concentration (% *w*/*w*)	pH	Type of Analyzed Fraction	Content (% *w*/*w*)	References
10 [37]	45	5	Pancreatin	2.5% (*w*/*v*)	8	YE	Proteins: 55.9% α-amino nitrogen: 1.8%	Bayarjargal et al., 2011 [37]
8 [76]	50	48	Flavourzyme	1% (*w*/*w*) E:S	7	YE	Proteins: 64.9% Carbohydrates: 26.9% Lipids: 0.8% α-amino nitrogen: 5.81% Main AA: Glu, Leu, Lys, Asp	Jung et al., 2011 [76]
100% (undiluted SBY) * [78]	60	5.5	Brauzyn	10% (*w*/*w*) E:S		YE	Proteins: 54.3% Carbohydrates: 40.9%	Marson et al., 2019 [78]
15 [79]	-	5	Alcalase	0.86% (*w*/*w*) E:S	7	YCW	Proteins: 23.33% β-glucan: 60.32% Lipids: 1.33%	Tam et al., 2013 [79]
5 [77]	50	24	Alcalase	-	7	YE	Proteins: 2.015%	Tsaroucha et al., 2022 [77]

* The yeast species used in the sample marked with ‘*’ was *Saccharomyces pastorianus*. The yeast strain used in the rest of the studies was *Saccharomyces cerevisiae*. Most of the enzyme concentrations are expressed with respect to the enzyme-to-substrate ratios (E:S).

**Table 3 ijms-25-12655-t003:** Summarization of cascade lysis experiments in correlation with the aim of the experiment.

Cascade Lysis Experiment *	Aim of the Experiment	References
Thermolysis, HPH Organic solvent treatment, EH	Extraction of β-glucan	Liu et al., 2008 [100]
Autolysis, HPH		Thammakiti et al., 2004 [101]
Alkaline treatment, HPH		Tian et al., 2019 [102]
HPH	Maximizing protein extraction	Barbieru et al., 2021 [103]
Autolysis, EH	Obtaining 5′-GMP-rich YE	Sombutyanuchit et al., 2001 [104]

* HPH–High-pressure homogenization, EH–Enzymatic hydrolysis.

**Table 4 ijms-25-12655-t004:** YCW applications overview.

Field	YCW Uses	YCW Effects	References
Pharmaceutical	Drug coating agent	Control of the release time of the active ingredient	Kasai et al., 2000 [117] Yuasa et al., 2000 [142] Yuasa et al., 2002 [143]
Materials science	Base matrix for biodegradable films	Forming sustainable materials for food packaging	Peltzer et al., 2018 [146]
Agricultural crop management	Bio stimulant	Enhances plant resistance to pathogens Exerts antifungal and antibacterial activity	Narusaka et al., 2015 [115]
Veterinary	Mycotoxin adsorbent	Improves survival rate and health of aflatoxin-infected broilers	Pereyra et al., 2018 [154] Liu et al., 2018 [155]

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
