# Peer review of "Spent Brewer’s Yeast Lysis Enables a Best Out of Waste Approach in the Beer Industry"

_ijms, 2024, doi:10.3390/ijms252312655_

Round 1
Reviewer 1 Report
Comments and Suggestions for Authors
The proposed manuscript focuses on an interesting and current problem, namely the exploitation of the biotechnological potential of spent yeast Saccharomyces cerevisiae from beer production. The authors review the state of the art in methods for lysing spent yeast. Similar reviews have been published in the literature, but the present manuscript describes the current knowledge on the advantages and disadvantages of each lysis technique. In addition, they provide information on the relevant experimental parameters of these techniques. The manuscript is a good contribution to the field, well-structured and scientifically sound. It is organized based on appropriate and adequate references, most of which have been published in the last 5 years. However, the manuscript needs more attention.
1. Authors should include more methods for yeast lysis in the manuscript such as freeze-thaw lysis, high-temperature lysis, chemical lysis, pulsed electric field, osmotic shock, etc.
2. They should add information on the target product for which the method is suitable. For example - Detergent lysis is suitable for protein release; Enzymatic lysis is suitable for extracting proteins; Osmotic shock is suitable for extracting sensitive intracellular products; etc.
3. In the section References there are literary sources with errors in the bibliography or incomplete information.
Author Response
Dear Reviewer,
We appreciate your valuable comments.
Therefore, we have revised the manuscript in accordance with your comments and suggestions. The changes we have made are detailed below. Please note that when the position of the modification made within the text is specified, reference is made to the line number in the revised manuscript saved with the track changes option.
Comment 1: Authors should include more methods for yeast lysis in the manuscript, such as freeze-thaw lysis, high-temperature lysis, chemical lysis, pulsed electric field, osmotic shock, etc.
Response 1: Thank you for pointing this out. We have included more lysis methods from up-to-date studies, which were reportedly applied to spent brewer’s yeast. Specifically, we have detailed the principles and main experimental results regarding high-temperature lysis, plasmolysis as an osmotic shock-based method (lines 257-293), and pulsed electric field lysis (lines 216-310). Some chemical treatments were also described (lines 201-212). Since our core objective was to focus solely on spent brewer’s yeast lysis, we aimed to select and detail the disruption methods that were specifically applied to this particular type of biomass.
Comment 2: They should add information on the target product for which the method is suitable. For example - Detergent lysis is suitable for protein release; Enzymatic lysis is suitable for extracting proteins; Osmotic shock is suitable for extracting sensitive intracellular products; etc
Response 2. Thank you for pointing this out. We have added information on the target products for which each of the described methods is suitable, specifically for autolysis (lines 150-153), high-temperature lysis (lines 232-239), pulsed electric field (lines 310-317), enzymatic hydrolysis (line 326), and ultrasonication (lines 455-456).
Comment 3: In the section References there are literary sources with errors in the bibliography or incomplete information.
Response 3. Thank you for pointing this out. We have revised the Reference section by correcting the errors and completing the source information.
Reviewer 2 Report
Comments and Suggestions for Authors
I am very grateful to you for the invitation to review the manuscript ijms-3283763 by Ciobanu and coauthors, titled "Spent brewer's yeast lysis enables a best out of waste approach in beer industry". This review highlights the needs for deeper investigation of molecular mechanisms to unleash the potential of spent brewer’s yeast extracts and cell walls to become an important source for a variety of bioactive compounds. The work is interesting but needs adjustments to improve the quality of the material.
Comments:
- Line 13: Highlight the main bioactive compounds present in the biomass.
- Lines 13-14: Include the amount generated annually.
- Line 20: Specify the main applications.
- Lines 31-35: Specify the key bioactive components of interest in this case.
- Line 44: Specify the amount generated annually.
- Lines 46-48: There are several studies in the literature exploring the lysis of spent brewer’s yeast to obtain various compounds for use in food and cosmetics (including Saccharomyces and Yarrowia). Please review and include them.
- Line 57: SBE?
- Spent brewer’s yeast cell lysis methods: Before this section, it is necessary to include an item detailing the characteristics of the biomass, including physicochemical composition, bioactive compounds present, and other important features for the extraction process.
- Line 95: Specify which endogenous enzymes.
- Line 143: Change “media. [20].” to “media [20].”.
- Line 275: Change “reported26.3%” to “reported 26.3%”.
- General: In each method, it is necessary to highlight the disadvantages inherent to any process, which should be clearly stated.
- Lines 282-292: This information should be included in a specific section for yeast characterization, which includes the presentation of the microorganism and its cell wall composition and structure (as mentioned earlier).
- Lines 450-451: The applications should be made clearer.
- 3.1. Applications and properties of YCW components: This section should be further explored, deeply highlighting the applications and potential of the components. There are several studies on the application, modification, and characterization of these components as bioactive and functional compounds that could be explored.
- Network bibliometric analysis of the main key-words: If this section is not thoroughly explored and expanded, it may be removed as it does not add significant information.
- Conclusion: It should be rewritten to present the main points regarding the use of spent brewer’s yeast and future perspectives in the field.
Author Response
Dear Reviewer,
We appreciate your valuable comments. Therefore, we have revised the manuscript in accordance with your comments and suggestions. The changes we have made are detailed below. Please note that when the position of the modification made within the text is specified, reference is made to the line number in the revised manuscript saved with the track changes option.
Comment 1: Line 13: Highlight the main bioactive compounds present in the biomass.
Response 1: Thank you for your comment. We have specified the main bioactive compounds from (spent brewer’s) yeast.
Comment 2: Lines 13-14: Include the amount generated annually.
Response 2: Thank you for your comment. We have included the estimated amount of spent brewer’s yeast generated worldwide every year (line 14), which was also further detailed (lines 47-50).
Comment 3: Line 20: Specify the main applications.
Response 3:
Thank you for your comment. We have specified the main applications and components of yeast cell walls (Line 21).
Comment 4: Lines 31-35: Specify the key bioactive components of interest in this case.
Response 4: Thank you for your comment. We have specified the key bioactive components in the Introduction section (lines: 41-42).
Comment 5: Line 44: Specify the amount generated annually.
Response 5: Thank you for your comment. We have detailed the main available data on the amount of spent brewer’s yeast that results globally every year from industrial beer production (lines 47-50).
Comment 6: Lines 46-48: Several studies in the literature explore the lysis of spent brewer’s yeast to obtain various compounds for use in food and cosmetics (including Saccharomyces and Yarrowia). Please review and include them.
Response 6: Thank you for your comment. We included the compounds recovered from SBY and their applications.
Comment 7: Line 57: SBE?
Response 7: Thank you for your comment. We have modified “SBE” to “SBY” (line: 69).
Comment 8: Spent brewer’s yeast cell lysis methods: Before this section, it is necessary to include an item detailing the characteristics of the biomass, including physicochemical composition, bioactive compounds present, and other important features for the extraction process.
Response 8: Thank you for your comment. We have included a section about Spent brewer’s yeast composition (lines 73-103).
Comment 9: Line 95: Specify which endogenous enzymes.
Response 9: Thank you for your comment. We have specified the endogenous enzymes (lines 137-138).
Comment 10: Line 143: Change “media. [20].” to “media [20].”.
Response 10: We have made the change (line: 189).
Comment 11: Line 275: Change “reported26.3%” to “reported 26.3%”.
Response 11: Thank you for your comment. We have made the change (line: 429).
Comment 12: General: In each method, it is necessary to highlight the disadvantages inherent to any process, which should be clearly stated.
Response 12: Thank you for your comment. We have specified the disadvantages of each method (lines 152-153, 232-234, 291-293,326-328, 452-453, 466-469, 510-512).
Comment 13: - Lines 282-292: This information should be included in a specific section for yeast characterization, which includes the presentation of the microorganism and its cell wall composition and structure (as mentioned earlier)
Response 13: Thank you for your comment. We have moved this paragraph to Section 2. Spent brewer’s yeast composition (lines: 74-83).
Comment 14: Lines 450-451: The applications should be made clearer.
Response 14: Thank you for your comment. We detailed several applications, Line 614-616.
Comment 15: 3.1. Applications and properties of YCW components: This section should be further explored, deeply highlighting the applications and potential of the components. There are several studies on the application, modification, and characterization of these components as bioactive and functional compounds that could be explored.
Response 15: Thank you for your comment. We have detailed the application of each yeast cell wall component separately, giving specific examples (lines 620-628, 632-635, 637-643, 644-648, 650-651, 658-664, 673-679, 683-691). We have also highlighted the application of whole yeast cell walls, highlighting their specific applications (lines 703-715, 719-723, 733-737 and Table 4).
Comment 16: Network bibliometric analysis of the main keywords: If this section is not thoroughly explored and expanded, it may be removed as it does not add significant information.
Response 16: Thank you for your comment. We have extended the bibliometric analysis to be informative (lines: 820-846).
Comment 17: Conclusion: It should be rewritten to present the main points regarding the use of spent brewer’s yeast and future perspectives in the field.
Response 17: The conclusions have been revised to include the main points regarding the uses of spent brewer’s yeast and future perspectives (lines 890-894).
Round 2
Reviewer 2 Report
Comments and Suggestions for Authors
Authors have improved the quality of the work.